# BLM-s/lE: A structured dataset of English spray-load verb alternations for testing generalization in LLMs

**Giuseppe Samo, Vivi Nastase, Chunyang Jiang, Paola Merlo**

Department of Linguistics

University of Geneva

{Giuseppe.Samo, Chunyang.Jiang, Paola.Merlo}@unige.ch

vivi.a.nastase@gmail.com

## Abstract

Current NLP models appear to be achieving performance comparable to human capabilities on well-established benchmarks. New benchmarks are now necessary to test deeper layers of understanding of natural languages by these models.

Blackbird's Language Matrices are a recently developed framework that draws inspiration from tests of human analytic intelligence. The BLM task has revealed that successful performances in previously studied linguistic problems do not yet stem from a deep understanding of the generative factors that define these problems.

In this study, we define a new BLM task for predicate-argument structure, and develop a structured dataset for its investigation, concentrating on the *spray-load* verb alternations in English, as a case study. The context sentences include one alternant from the *spray-load* alternation and the target sentence is the other alternant, to be chosen among a minimally contrastive and adversarial set of answers. We describe the generation process of the dataset and the reasoning behind the generating rules. The dataset aims to facilitate investigations into how verb information is encoded in sentence embeddings and how models generalize to the complex properties of argument structures.

Benchmarking experiments conducted on the dataset and qualitative error analysis on the answer set reveal the inherent challenges associated with the problem even for current high-performing representations.

## 1 Introduction

In recent years, fuelled by the rise of deep learning methods, NLP models have undergone a remarkable transformation, becoming exceptionally powerful and achieving performance comparable to human capabilities on well-established benchmarks (Wang et al., 2019; Rajpurkar et al., 2018).

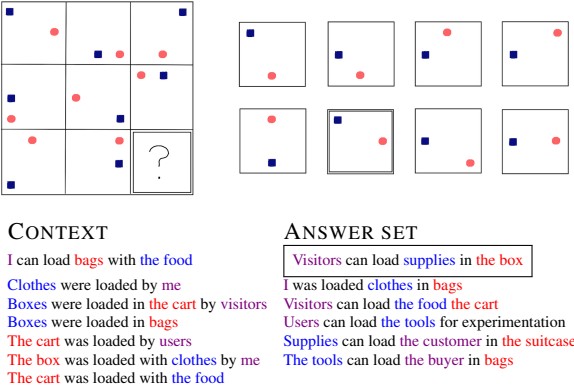

Figure 1: Upper panel: An example of Raven's Progressive Matrices. The task is to identify a missing element in a visual pattern from a list of candidate options (on the right) based on the given matrix (on the left). The matrix is created using specific rules for the placement of visual elements (e.g. the red dot and the blue square). Lower panel: An example of the Blackbird Language Matrices. The task is to identify a missing element in a linguistic pattern from a list of candidate options (on the right) based on the provided matrix (on the left). The matrix of sentences is constructed using rules of syntactic and semantic nature.

New benchmarks are now necessary to investigate deeper layers of comprehension of natural languages (Ruder, 2021).

A recently developed framework is the Blackbird's Language Matrices (BLM; Merlo et al. 2022; Merlo 2023a,b). This approach draws inspiration from tests of analytic intelligence, specifically the task of detecting visual patterns (Raven's Progressive Matrices (RPM), Raven 1938). The BLM task has revealed that correct predictions in previously studied linguistic problems do not yet stem from a deep understanding of the generative factors that define these problems. For example, previous work has demonstrated that RNN can predict the correct verb form with high accuracy based on the grammatical number of the subject (Linzen et al. 2016; Gulordava et al. 2018; see Linzen and Baroni 2021

for an overview). A more in depth exploration using a BLM task showed that despite the high results of previous studies, detecting the factors relevant to the subject-verb agreement rule is a more challenging problem (An et al., 2023).

In this paper, we adopt BLMs to investigate if current models reach deeper understanding of a core property of language, a verb's argument structure. The argument structure of a verb, the mapping of semantic roles to grammatical functions frames, is a core building block in claiming understanding of any sentence. Argument structure learning finds a particularly difficult case in those verbs that allow alternations, the multiple mappings of semantic roles to grammatical function and subcategorisation frame for the same verb. We exemplify this problem by studying a linguistic phenomenon in English, known as the *spray/load* alternation (Rappaport and Levin, 1988; Levin, 1993). Analogously to the subject-verb agreement problem, we revisit the recent results that typologies of neural architectures handle a binary classification of this phenomenon with ease (Kann et al., 2019; Yi et al., 2022).

The syntactic behaviour involves an alternation in which a selected class of verbs combines three arguments to describe an event where an AGENT causes motion of a THEME to a LOC(ATION). Both THEME and LOC can function as the syntactic direct objects of the structure, resulting in two possible alternating configurations. In one alternate, THEME immediately follows the verb, and LOC is introduced by a preposition (*The student sprays the paint onto the wall*). In the other alternate, LOC follows the verb, and THEME is introduced by a preposition (*The student sprays the wall with the paint*).

We develop BLM data that expose the model to syntactic properties of the arguments THEME and LOC, implicitly showing that the two alternates share common properties. To learn the alternation then, the model must be able to generalize from these shared properties.

We present the generation process of *spray/load* BLMs in Section 2 and the generation of the associated dataset for *spray/load* alternation in Section 3. Experiments with different types of architectures are presented in Section 4. The code and the data are available here: https://github.com/CLCL-Geneva/BLM-SNFDisentangling.

The contribution of this paper are three-fold.

- We define a new BLM matrix template, applied to *spray/load* verbs, which can also be of general interest as a blueprint for many alternations.

- We create a large data set on this phenomenon, which supports probing of neural network models.

- We provide a qualitative error analysis of the learned knowledge of core syntactic and semantic properties for different neural network models. This analysis shows that the learned properties in more complex configurations are limited to the syntactic aspects of the argument structure.

## 2   BLM-s/lE generation process

Task-solvers in RPM identify patterns based on the available information to identify the missing piece that completes the matrix (Kisielewska et al., 2016). Similarly, BLMs correspond to a problems where only one answer satisfies the constraints defined by the given contexts. In a BLM problem, a linguistic phenomenon is presented as an incomplete sequence of sentences (*context*), deliberately designed to follow given linguistic rules and given linguistic properties.

| | CONTEXT |
|---|---|
| 1 | Alternate 1 |
| 2 | Passivized THEME with overt AGENT |
| 3 | Passivized THEME with overt AGENT and LOC |
| 4 | Passivized THEME with overt LOC |
| 5 | Passivized LOC with overt AGENT |
| 6 | Passivized LOC with overt AGENT and THEME |
| 7 | Passivized LOC with overt THEME |
| ? | Alternate 2 |

Figure 2: Contexts template for BLM-s/lE

The task is to identify the missing element that continues the context pattern based on the observed relationships. The missing element is the only correct answer that represents the solution to the context pattern, among many elements in a carefully curated answer set consisting of minimally differing sentences (Merlo et al. 2023a,b).

### 2.1   The context indicators of underlying rules

While the *spray/load* alternation is the expression of complex syntactic and semantic interactions in a verb argument structure, a visible surface indicator of these properties is the use of the passive. Passivizability in English is commonly assumed to

| | CONTEXT | | | |
|---|---|---|---|---|
| 1 | NP-Agent | Verb | NP-Loc | PP-Theme |
| 2 | NP-Theme | VerbPass | PP-Agent | |
| 3 | NP-Theme | VerbPass | PP-Loc | PP-Agent |
| 4 | NP-Theme | VerbPass | PP-Loc | |
| 5 | NP-Loc | VerbPass | PP-Agent | |
| 6 | NP-Loc | VerbPass | PP-Theme | PP-Agent |
| 7 | NP-Loc | VerbPass | PP-Theme | |
| ?8 | NP-Agent | Verb | NP-Theme | PP-Loc |

| | ANSWERS | |
|---|---|---|
| 1 | **NP-Agent Verb NP-Theme PP-Loc** | CORRECT |
| 2 | NP-Agent *VerbPass NP-Theme PP-Loc | AGENTACT |
| 3 | NP-Agent Verb NP-Theme *NP-Loc | ALT |
| 4 | NP-Agent Verb *PP-Theme PP-Loc | ALT |
| 5 | NP-Agent Verb *[NP-Theme PP-Loc] | NOEMB |
| 6 | NP-Agent Verb NP-Theme *PP-Loc | LEXPREP |
| 7 | *NP-Theme Verb NP-Agent PP-Loc | SSM |
| 8 | *NP-Loc Verb NP-Agent PP-Theme | SSM |
| 9 | *NP-Theme Verb NP-Loc PP-Agent | AASSM |

Figure 3: BLM context and answers for the spray/load alternation from AGENT-LOC-THEME to AGENT-THEME-LOC. CR = Corrupted rule(s), * = locus of the rule corruption, angled brackets = syntactic embedding.

| | CONTEXT | | | |
|---|---|---|---|---|
| 1 | NP-Agent | Verb | NP-Theme | PP-Loc |
| 2 | NP-Theme | VerbPass | PP-Agent | |
| 3 | NP-Theme | VerbPass | PP-Loc | PP-Agent |
| 4 | NP-Theme | VerbPass | PP-Loc | |
| 5 | NP-Loc | VerbPass | PP-Agent | |
| 6 | NP-Loc | VerbPass | PP-Theme | PP-Agent |
| 7 | NP-Loc | VerbPass | PP-Theme | |
| ?8 | NP-Agent | Verb | NP-Loc | PP-Theme |

| | ANSWERS | |
|---|---|---|
| 1 | **NP-Agent Verb NP-Loc PP-Theme** | CORRECT |
| 2 | NP-Agent *VerbPass NP-Loc PP-Theme | AGENTACT |
| 3 | NP-Agent Verb NP-Loc *NP-Theme | ALT |
| 4 | NP-Agent Verb *PP-Loc PP-Theme | ALT |
| 5 | NP-Agent Verb *[NP-Loc PP-Theme] | NOEMB |
| 6 | NP-Agent Verb NP-Loc *PP-Theme | LEXPREP |
| 7 | *NP-Loc Verb NP-Agent PP-Theme | SSM |
| 8 | *NP-Theme Verb NP-Agent PP-Loc | SSM |
| 9 | *NP-Loc Verb NP-Theme PP-Agent | AASSM |

Figure 4: BLM context and answers for the spray/load alternation from AGENT-THEME-LOC to AGENT-LOC-THEME. CR = Corrupted rule(s), * = locus of the rule corruption, angled brackets = syntactic embedding.

reflect the argument structure of the active sentence, specifically its transitivity (Hopper and Thompson, 1980). In the *spray/load* verb class, the THEME and LOC can be syntactically transformed into the subject of a passive structure (e.g., *paint was sprayed onto the wall* vs. *the wall was sprayed with paint*, as they both share direct object properties (D'Elia, 2016, 156-159).

The matrix then is structured to provide one of the two alternates in the alternation, and provide shared properties (passivization of THEME and LOC) and to interrogate the model to output the other alternate as the missing, target sentence. This structure is illustrated in Figure 2, and represents the BLM for the *spray-load* alternation in English, referred to as BLM-s/lE.

## 2.2 The matrix templates

Each sentence can be described in terms of one distribution-of-three-values rule, governing the semantic roles (Agent, Theme, Locative), and two distribution-of-two-values rules governing syntactic types (nominal phrase NP vs. prepositional phrases PP) and the mood of the verb, whether active (Verb) or passive (VerbPass). We created two templates, one for each of the two alternates, as shown in Figures 4 and Figure 3.

We call ALT-ATL data, the data produced from the matrix in Figure 3, and ATL-ALT data those produced from the matrix in Figure 4. Examples are discussed in section 3.

## 2.3 The contrastive answer set

The target sentence is to be chosen from a set of candidates that exhibit minimal differences. The semantic-syntactic mapping of the alternation can be decomposed into a set of smaller patterns that describe the sentences in the alternation and that can be violated to construct incorrect answers. We list them below: subpatterns 1-3 govern the correct learning of the syntactic form of the sentence; subpattern 4 governs the proper lexical selection and pattern of type 5 governs the proper mapping syntax-semantic mapping (which can be expressed in many ways).[1]

1. Agent position in an active sentence (AGENTACT): If the voice of the sentence is active, the agent is an NP in subject position .

2. Preserve to the alternation pattern (ALT): The verb is followed by a NP and a PP.

3. No Embedding (NOEMB): The PP following the NP is not embedded in it.

4. The lexical choice of the preposition (LEXPREP): When PPs, arguments are introduced by given prepositions (e.g. *onto* the wall vs. *under* the wall).

5. Syntax-Semantic Mapping (SSM): the order of the constituents and their role is fixed (e.g. a Theme cannot be in an Agent position in active mood).

---

[1]Multiple rules can be corrupted at the same time. For example, AGENTACT and SSM can be both violated at the same time (e.g. *The wall sprayed paint with the girl*). We label such violation as AASSM.

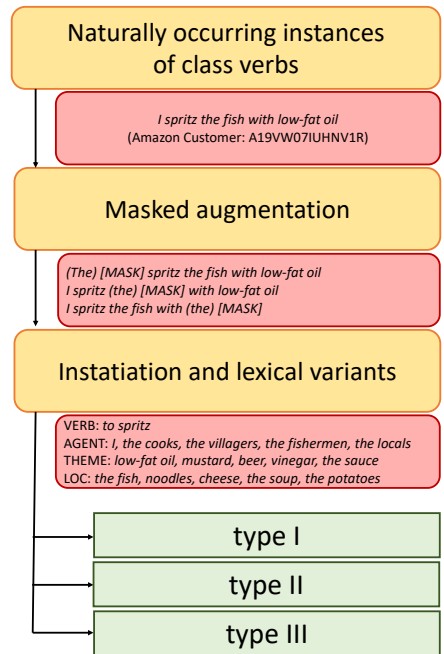

Figure 5: Pipeline process. In red, an example of the steps with the verb *to spritz* is provided.

The structure of the answer set is presented in Figures 3 and 4 for each context. Relevant examples are given in section 3.

## 3 Creating the dataset

To instantiate the templates, we follow a pipeline process comprising several steps.

1. Find relevant verbs in naturally occurring seed sentences.

2. Augment these sentences to create a large dataset.

3. Recombine the seed and augmented items to further augment the size of the dataset in a controlled way.

Each step is discussed in the following paragraphs. Figure 5 illustrates the workflow.

**Natural instances of class verbs** Dedicated rich repositories grouping verbs into semantically coherent classes based on shared syntactic behavior exist (Stowe et al., 2021). We retrieved the lexical verbs belonging to the same class as *spray*[2] and the verb *load* from VERBNET (Kipper Schuler

[2] https://verbs.colorado.edu/verb-index/vn/spray-9.7.php#spray-9.7-1.

2005) to identify a set of 30 verbs that undergo the *spray/load* alternation.[3]

The data generation process begins by selecting a naturally occurring example for each verb in the class. To accomplish this, we use the SPIKE platform[4] (Shlain et al., 2020), which offers a collection of English corpora from various text genres (encyclopedic entries, internet reviews, among others). We favoured extracting natural contexts, and we prioritized examples obtained from the Amazon Reviews subcorpus. We performed the search of different inflected forms of the verbs and analysed manually the naturally occurring examples in the output. For example, the search for different inflected forms of the verb *spritz* allowed us to retrieve a sentence like *I spritz the fish with low-fat oil* (Amazon Customer; A19VW07IUHNV1R).

We removed non-target examples (e.g. those cases in which the PP does not represent an argument of the verb, but it is embedded in the nominal phrase, as in *This bulb scatters dim light with a green hue around*; ID: AMG Enthusiast; AI9JTX5ZX0OJG). Care was also taken to vary the types of sentences as much as possible (e.g. imperatives, relatives) and pronominal entities (e.g., to counter the fact that a good portion of Amazon reviews involving these verbs are indeed in first person singular).

**Masked augmentation** The extracted examples were then used in a fill-mask task. Every naturally occurring example (seed) had in all six different maskings. Each of the three arguments (AGENT, THEME and LOC) was masked in two contexts, one aiming at eliciting an indefinite noun (without the article, e.g. *I spritz [MASK] with low fat oil*) and the other at eliciting a definite noun (preceded by the definite article *the*, eg. *I spritz the [MASK] with low fat oil*). We used the DistilBERT uncased model (Sanh et al., 2019), as we want to use a different language model for data augmentation from the one we will use for learning (section 4) to avoid bias. From the proposed items, we manually selected five elements that, when combined, produced grammatically and semantically acceptable sentences (for example, they belonged to a similar semantic field as the seed). We also took definiteness into account. Pronouns were limited to those

[3] In alphabetical order: *baste, brush, drizzle, hang, load, plaster, pump, rub, scatter, seed, sew, shower, smear, smudge, sow, spatter, splash, splatter, spray, spread, sprinkle, spritz, spurt, squirt, stick, strew, string, swab, swash, wrap.*

[4] https://spike.apps.allenai.org/datasets

CONTEXT

| | |
|---|---|
| 1 | The buyer can load the tools in bags. |
| 2 | The tools were loaded by the buyer |
| 3 | The tools were loaded in bags by the buyer |
| 4 | The tools were loaded in bags |
| 5 | Bags were loaded by the buyer |
| 6 | Bags were loaded with the tools by the buyer |
| 7 | Bags were loaded with the tools |
| 8 | ??? |

Figure 6: Example of Type I context sentences.

CONTEXT

| | |
|---|---|
| 1 | I pumped the pipes with regular air |
| 2 | The grain was sown by the monks |
| 3 | Regular air was pumped into the stove by me |
| 4 | The contents were stuck in a box |
| 5 | The suitcase was loaded by the buyer |
| 6 | The dragon world was scattered with monsters by the wizard |
| 7 | The surface was splattered with the stuffing |
| 8 | ??? |

ANSWERS

| | |
|---|---|
| 1 | **The tomatoes splatter water all over the grill** |
| 2 | The scientists were seeded sulfur into the earth |
| 3 | Farmers have to squirt a tiny bit of oil everything |
| 4 | My child smeared with the flour all over the room |
| 5 | People baste different types of marinades of seafood |
| 6 | I brushed mascara under the surface |
| 7 | The yummy almond butter spread someone all over the pastry |
| 8 | The stuff sticks Bob in the truck |
| 9 | The cart can load the supplies in the buyer |

Figure 7: Example of Type III context sentences and answer set.

present in naturally occurring examples.

Two of the authors individually validated each item. A consensus decision was then reached for each item. By the end of this process, then, each of the 30 verbs was related to five AGENTS, five THEME and five LOC. We refer to this group of sentences as the *lexical seed set*.

**Instantiation and lexical variants** The constituents belonging to the lexical seed set were merged together to build templates and answer sets. We constructed three types of contexts and answer sets, each of increasing lexical difficulty. Type I refers to problems generated directly with seed input segments and their variations: the same lexical elements for the verb, AGENT, THEME, and LOC are found for every sentence in each context. The combination of five arguments (125) x 30 verbs creates 3750 contexts/answers sets. In Type II the templates are built with the same verb, but the arguments vary lexically. Finally, Type III results from a fully random reshuffle of Type II: for each context and answer set, each sentence varies in terms

of lexical entries of the arguments and verb. Type II and Type III comprise 15,000 contexts/answers sets. Examples of type I context and type III are shown in Figures 6 and 7. Remaining examples are shown in the appendix.

Each of the three subsets of datasets is split 90:10 into train and test subsets, which are provided with the data. 20% of the train data is used for development. Section 4 presents the experimental settings.

## 4 Experiment

We present two baselines to investigate the challenge of detecting the underlying patterns of complex syntactic-semantic mapping in transformer-based sentence embeddings (Hewitt and Manning, 2019; Yi et al., 2022). Our BLMs for the *spray/load* alternation are a case study of this kind of problem. To provide a benchmark for the task, we choose architectures that are simple, but effective – a three-layer feed-forward neural network, and a three-layer convolutional neural network – as proven by their numerous applications in NLP. They are described in detail in section 4.2.

### 4.1 Data

We use the BLM data described above, and 1x768 sentence embeddings generated using RoBERTa (Liu et al., 2019) and Electra (Clark et al., 2020) pretrained models.[5] For each sentence in a BLM instance, we use the embedding of the [CLS] /  token as its representation.

### 4.2 Baseline systems

We aim to detect patterns that encode linguistic information concerning verb alternations in sequences of sentences. We choose two baseline systems – a feed-forward neural network (FFNN) and a convolutional neural network (CNN). Each has characteristics that will allow it to find patterns – either more localized (the CNN) or more distributed (the FFNN) – in the input sequences.

The FFNN receives input as a concatenation of sentence embeddings in a sequence, with a size of 7 x 768. This input is then processed through 3 fully connected layers, which progressively compress the input size (7 x 768 $\xrightarrow{layer1}$ 3.5 x 768 $\xrightarrow{layer2}$ 3.5 x 768 $\xrightarrow{layer3}$ 768) to obtain the size of a sentence representation. The FFNN's interconnected layers

---

[5]RoBERTa: *xml-roberta-base*, Electra: *google/electra-base-discriminator*

enable it to capture patterns that are distributed throughout the entire input vector.

The CNN takes as input an array of embeddings with a size of (7 x 768). This input undergoes three consecutive layers of 2-dimensional convolutions, where each convolutional layer uses a kernel size of (3x3) and a stride of 1, without dilation. The resulting output from the convolutional process is then passed through a fully connected layer, which compresses it to the size of the sentence representation (768). By using a kernel size of (3x3), stride=1, and no dilation, this configuration emphasizes the detection of localized patterns within the sentence sequence array.

Both networks output a vector representing the sentence embedding of the correct answer. The objective of learning is to maximize the probability of selecting the correct answer from a set of candidate answers. To achieve this, we employ the max-margin loss function, considering that the incorrect answers in the answer set are intentionally designed to have minimal differences from the correct answer. This loss function combines the distances between the predicted answer and both the correct and incorrect answers. Initially, we calculate a score for each candidate answer's embedding $e_i$ in the answer set $\mathcal{A}$ with respect to the predicted sentence embedding $e_{pred}$. This score is determined by the cosine of the angle between the respective vectors:

$$score(e_i, e_{pred}) = cos(e_i, e_{pred})$$

The loss function incorporates the max-margin concept, taking into account the difference between the score of the correct answer $e_c$ and each of the incorrect answers $e_i$:

$$loss_a = \sum_{e_i}[1 - score(e_c, e_{pred}) + score(e_i, e_{pred})]^+$$

During prediction, the answer with the highest score value from the candidate set is selected as the correct answer.[6]

## 4.3 Results

We performed experiments with RoBERTa sentence embeddings using both baselines. Because

the FFNN perfomed much better, we then performed experiments with Electra sentence embeddings only with the FFNN.

Figure 13 shows the heatmaps of F1 scores for ATL-ALT (calculated as averages over 5 runs). Heatmaps of ALT-ATL can be found in the appendix. On the left side of the panel are results on all data, on the right side are results where type II and type III training data are sampled to a sample size comparable to type I.

We did not detect any particular asymmetries in terms of performance between the two groups, suggesting that there is no a favoured alternate. While the results for all configurations are quite high, the most difficult setup is when the model is trained on data with the least amount of lexical variation (type I), and tested in the data with the highest lexical variation (type III). However, when using Electra sentence embeddings, even this setup has a performance of F1 $> 0.80$, showing that in Electra sentence embeddings, lexical variation is not a deterring factor neither in finding relevant patterns for verb alternations, nor in applying them.

Overall, Electra performs better, confirming the results in Yi et al. (2022).

## 4.4 Error Analysis

Error analysis can help determine which are the more problematic patterns to detect in sentence embeddings. We perform this analysis for the more challenging setting: models trained on type III data, and tested on all the others. The corresponding plots are presented in Figures 9 and 10.

**Errors common to both groups** Across models, the highest error is mainly the SSM (e.g., *The wall sprayed the girl with the paint*), which represents an argument structure mistake, as the mapping from semantic roles to syntactic functions is mistaken. This is indeed a difficult aspect of the alternation to learn, as the surface syntactic structure is correct, and the incorrect alternatives can be detected only if the underlying lexical semantic rule is mastered (Agent=subject, theme=object), namely, if the model has somehow acquired knowledge of semantic macroroles, together with world knowledge about what can be a Agent or a Theme for the verb *spray*, which corresponds to semantic microroles.

ALT (e.g., *The girl sprayed the wall the paint*; *the girl sprayed over the wall with the paint*) is also a very common error observed when testing

[6]All systems used a learning rate of 0.001 and Adam optimizer, and batch size 100. The training was done for 120 epochs. The experiments were run on an HP PAIR Workstation Z4 G4 MT, with435 an Intel Xeon W-2255 processor, 64G RAM, and a MSI GeForce RTX 3090 VENTUS 3X OC 24G GDDR6X GPU.

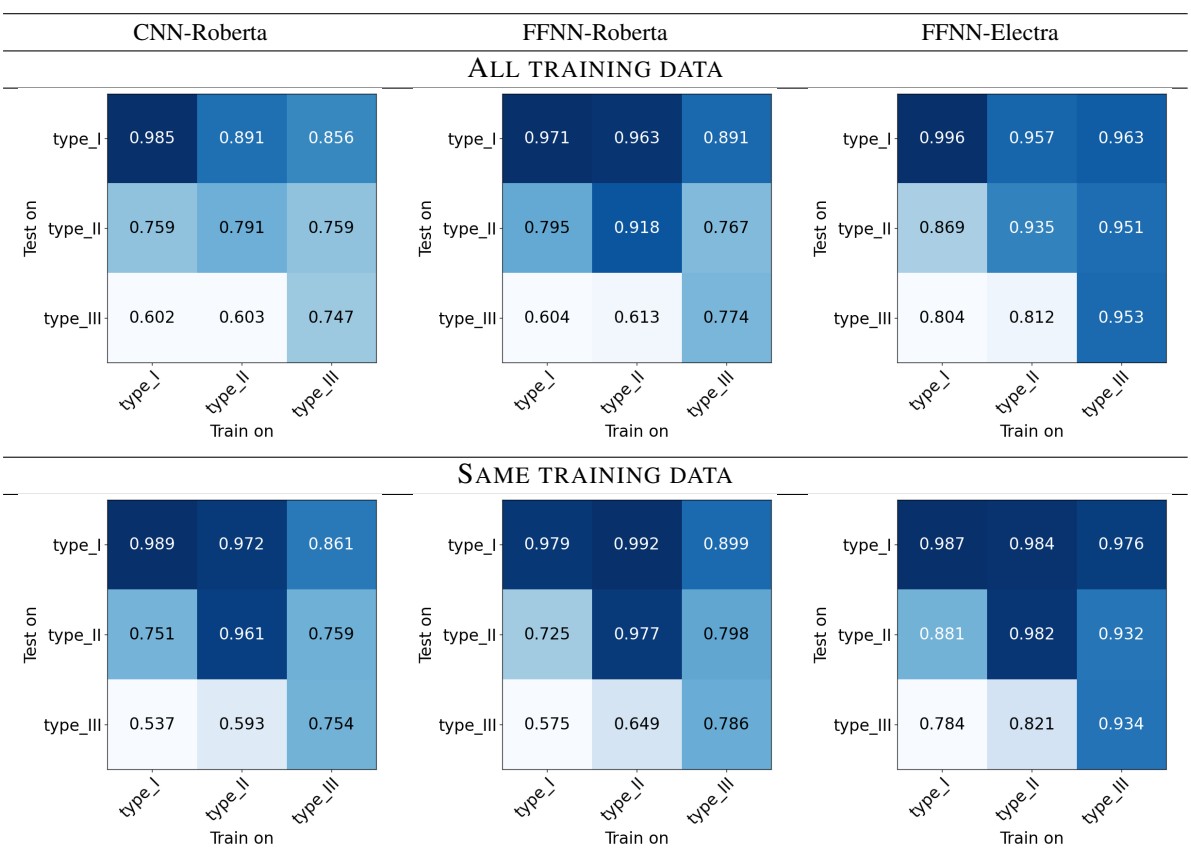

Figure 8: F1 results for ATL-ALT (given AGENT-THEME-LOC guess AGENT-LOC-THEME).

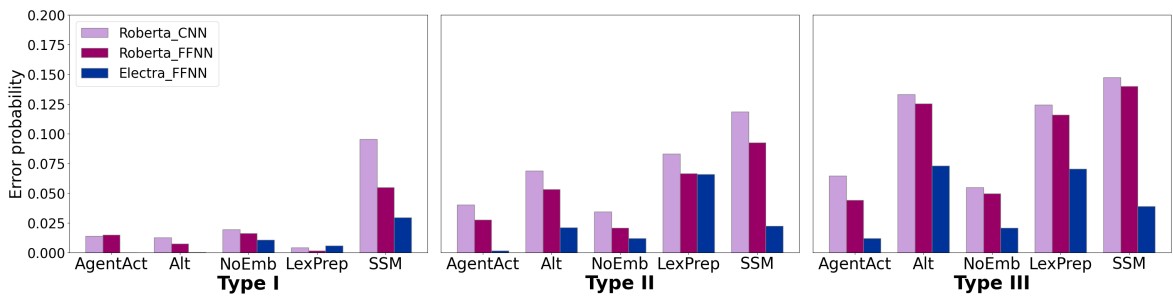

Figure 9: Error analysis for ATL-ALT (given AGENT-THEME-LOC predict AGENT-LOC-THEME. For example, given *spray the paint onto the wall* predict *spray the wall with paint*, aggregated trainings)

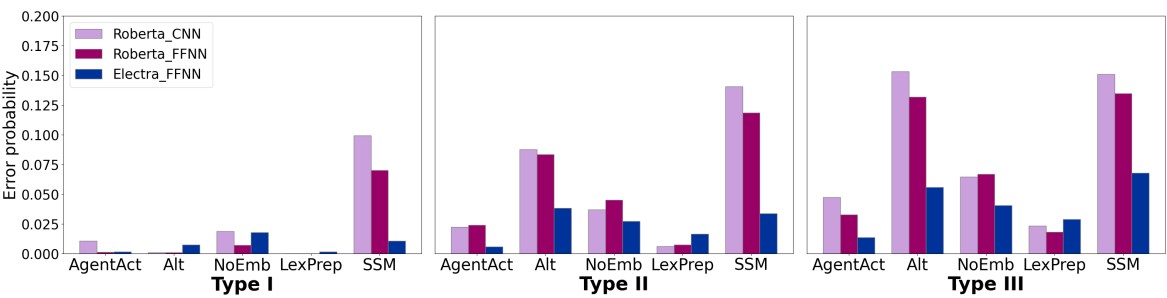

Figure 10: Error analysis for ALT-ATL (given AGENT-LOC-THEME predict AGENT-THEME-LOC. For example, given *spray the wall with paint* predict *spray the paint onto the wall*, aggregated trainings)

the model on type III data. This means that the word order of semantic roles of the alternations are learned, but the syntax is not.

AGENTACT mistakes are also frequent (*The girl was sprayed the wall with the paint*). Here the sentence is ungrammatical, but it is in the passive voice, like the majority of sentences in the context, so it is close to the context by structural similarity. This suggests that the underlying rules are not decomposed, but learnt by analogy and similarity in a global space.

NOEMB is less problematic (*the girl sprayed the paint for the room*). This contrasts with the high number of mistakes in lexical choice of the preposition in some cases. LEXPREP seems problematic for type III of group ATL-ALT (*the girl sprayed the paint under the wall*). This mistakes shows that the models can detect correct from incorrect sentences –this is afterall a plausible sentence– but it shows that the general pattern of the alternation has not been identified at all.

**Errors most prominent in one group**    There are asymmetries between the two groups regarding specific types of errors. The occurrence of errors related to the LexPrep rule is more prevalent when the target sentence follows the AGENT-LOC-THEME structure, for example, *the girl sprayed the wall under the paint*. The limited number of errors of this kind in the other group suggests that the encoding of the preposition information for the locative element, when it is the target, is well preserved (e.g. *the girl sprayed paint under the wall*).

As mentioned earlier, ALT is a highly frequent error observed during testing on type III. Asymmetries emerge in the nature of these errors, with two distinct types occurring. The first type involves the verb being followed by two noun phrases (*the girl sprayed the wall the paint*), while the second type involves the verb being followed by two prepositional phrases (*the girl sprayed onto the wall with the paint*). Although the latter error is more prominent across the three models in both groups, there are distributional differences between the two groups. Moreover, in the ATL-ALT, a clear difference in performance can be observed between Roberta and Electra models.

## 4.5    Discussion

According to the error analysis, the observed patterns of successful generalization predominantly concern syntactic aspects. The semantic mapping of arguments continues to pose significant challenges. In particular, we observed that the highest number of errors is related to the mapping between semantic roles and syntactic functions. We believe that mastering this aspect of the alternation proves to be challenging because the surface syntactic structure of incorrect alternatives is mostly correct, and identifying the correct answer requires that the underlying lexical semantic properties be identified in the distributed and continuous sentence representations, which the baseline models are not able to do.

## 5    Related work

The current paper does not have a direct comparison, since it is the first proposal of a dataset for a verb alternation adopting a BLM scheme. It is directly connected to the works on BLM on agreement, studies on verb classification and syntactic rules and, partially, on structured datasets adopted in computer vision (Wang and Su, 2015; Zhang et al., 2019).

**BLM for agreement**    The proposed structured dataset enriches the typology of linguistic phenomena that can be tested with BLMs. Previous studies focused on investigating agreement patterns between the subject and verb in French (An et al., 2023; Merlo et al., 2022). One of the difficulties of the experiments was provided by the increasing number of attractors between the subject and the verb, which could potentially interfere with the detection of the agreement relation between them. Another difficulty was that the agreement patterns followed themselves a global pattern in the matrix that had to be identified. Two exploratory baselines showed that despite the simplicity of the phenomenon, deep learning system do not manage to identify the underlying rules.

**Verb Alternations**    Previous work has focussed on understanding the automatic learning of verb alternations in terms of syntactic and semantic properties of the verbs and their argument structures (allowing, or disallowing verb alternation) (see Majewska and Korhonen 2023 and reference therein). These properties have been explored in relation to their representation in Large Language Models (Kann et al., 2019; Yi et al., 2022), encompassing various dimensions of performance across different models. In particular, Yi et al. (2022) suggest that LLMs with contextual embeddings encode lin-

guistic information on verb alternation classes, at both the word and sentence levels. Specifically, for the *spray/load* alternation, Yi et al. (2022) probed whether a verb entered this class by building two binary classifiers. The classifier predicts whether a verb can participate in both syntactic frames of the alternation (frame intended as syntactic environment, see also Kann et al. (2019)). The binary classification task – done based on token embeddings – makes the targeted phenomenon explicit, but does not provide insights into how this linguistic property of verbs is encoded, or how it interacts with the verb's arguments.

## 6 Conclusion

In this paper, we have introduced BLM-s/lE, a BLM-structured dataset for the English *spray/load* verb alternation. This is a complex case study for the larger problem of predicate-argument structure.

The dataset comprises two groups of templates. Each group presents one of the alternants, while the other alternant needs to be inferred based on the information provided by the context of the BLM. The answer set is designed to exhibit minimal superficial variations from the correct answer but significant differences in the underlying structure. Through error analysis, we gain a better understanding of the types of information that baseline models manage to learn, and what aspects instead will require specifically developed models and architectures.

Our findings indicate that, on the one hand, curated and structured datasets, such as the one presented in this study, have the potential to lead to articulated understanding of learning. On the other hand, we conducted experiments using two different architectures, and the results highlight the usefulness of the task, whose solution requires detecting underlying linguistic properties and rules. Overall, these findings provide further evidence supporting the use of our BLMs as an innovative benchmark to reach better qualitative understanding of learning in current neural network models.

## Limitations

**Additional alternation phenomena**  As this work serves as an initial blueprint, our decision has been to concentrate solely on a single verb alternation. However, it will be interesting for the future to aim for a broader spectrum of verb alternations and in multiple languages.

**Human upper bound**  We do not have, at this stage, comparable results with human behaviour, due to the type and size of the task.

## Ethics Statement

To the best of our knowledge, there are no ethics concerns with this paper.

## Acknowledgements

We gratefully acknowledge the partial support of this work by the Swiss National Science Foundation, through grants #51NF40_180888 (NCCR Evolving Language) and SNF Advanced grant TMAG-1_209426 to PM.

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

# A  Appendix

## A.1  Answer set type I

TYPE I

| EXAMPLE OF CONTEXT |
| --- |
| The buyer can load the tools in bags. |
| The tools were loaded by the buyer |
| The tools were loaded in bags by the buyer |
| The tools were loaded in bags |
| Bags were loaded by the buyer |
| Bags were loaded with the tools by the buyer |
| Bags were loaded with the tools |
| ??? |

| EXAMPLE OF ANSWERS |
| --- |
| **The buyer can load bags with the tools** |
| The buyer was loaded bags with the tools |
| The buyer can load bags the tools |
| The buyer can load in bags with the tools |
| The buyer can load bags on sale |
| The buyer can load bags under the tools |
| Bags can load the buyer with the tools |
| The tools can load the buyer in bags |
| Bags can load the tools in the buyer |

Figure 11: Example of Type I context sentences and answer set.

## A.2  Type II: example of context sentences and answer set

TYPE II

| EXAMPLE OF CONTEXT |
| --- |
| I can load bags with the food |
| Clothes were loaded by me |
| Packages were loaded in the cart by visitors |
| Packages were loaded in bags |
| The cart was loaded by users |
| The box was loaded with clothes by me |
| The cart was loaded with the food |
| ??? |

| EXAMPLE OF ANSWERS |
| --- |
| **Visitors can load supplies in the box** |
| I was loaded clothes in bags |
| Visitors can load the food the cart |
| Visitors can load with the food in bags |
| Users can load the tools for experimentation |
| Users can load the food under bags |
| Supplies can load the customer in the suitcase |
| The tools can load the buyer in bags |
| The cart can load the supplies in the buyer |

Figure 12: Example of Type II context sentences and answer set.

## A.3 Detailed F1 scores

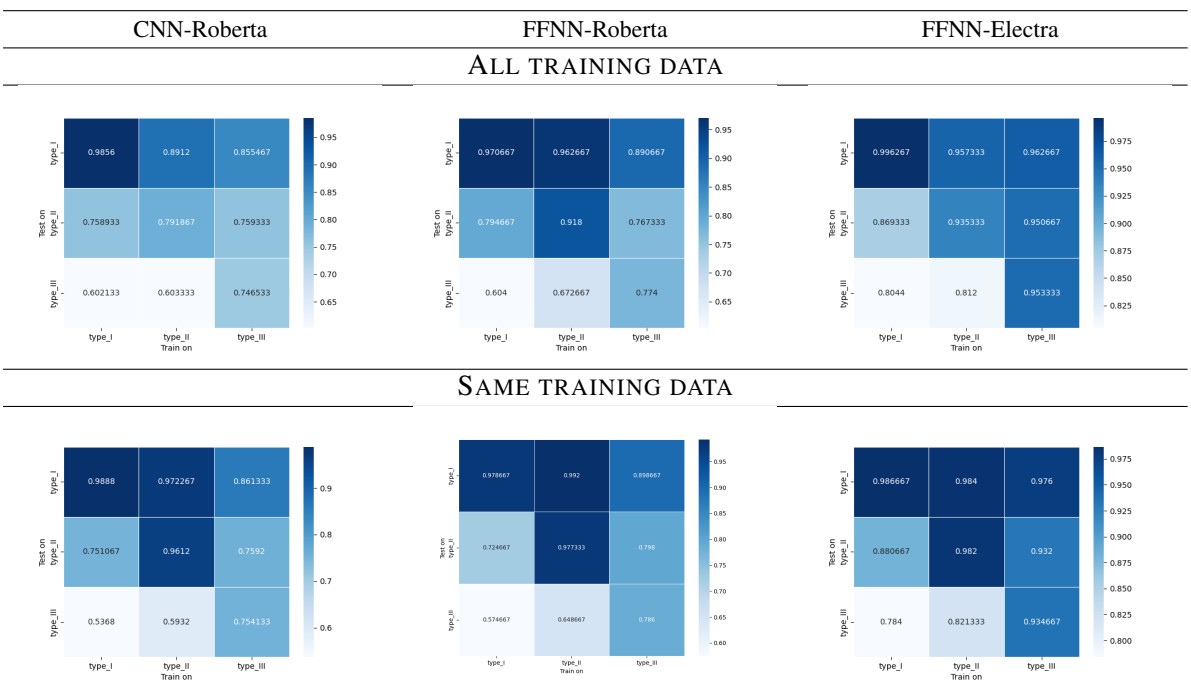

Figure 13: F1 results for ATL-ALT group (given AGENT-THEME-LOC predict AGENT-LOC-THEME).).

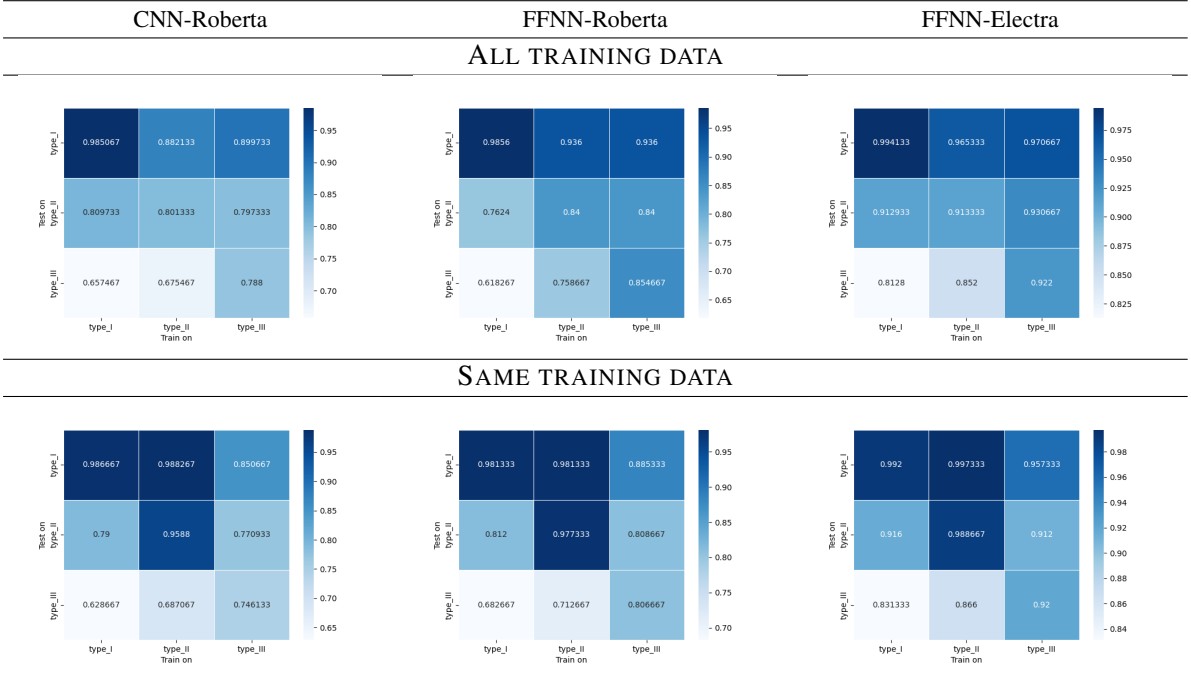

Figure 14: F1 results for ALT-ATL group (given AGENT-LOC-THEME predict AGENT-THEME-LOC)..