# OpenReview forum: "BLM-s/lE: A structured dataset of English spray-load verb alternations for testing generalization in LLMs"
_EMNLP/2023/Conference — EMNLP 2023 Findings_

### Official Review · Reviewer_YHa3 · 2023-08-04

**Soundness:** 3

**Excitement:**

2: Mediocre: This paper makes marginal contributions (vs non-contemporaneous work), so I would rather not see it in the conference.

**Paper Topic And Main Contributions:**

This paper presents a new dataset for the spray/load verb alternations in English. The dataset aims to facilitate investigation of LLMs understanding of argument structure of a verb.  The authors also present a qualitative analysis of the errors produced by the 3 models considered.


**Questions For The Authors:**

a) With growing number of available datasets for probing understanding of LLMs, how do the authors foresee extending and maintaining this dataset? If such a test were to be used for benchmarks, how might they be extended for other languages?


**Reasons To Accept:**

The authors note the need for new benchmarks to probe language understanding of LLMs. To this end they present a case of alternations of verb argument structure for the English spray/load. Leveraging linguistic insight to examine the 'understanding' of LLMs is crafty.


**Reasons To Reject:**

The paper would benefit from further proof reading. There are instances where it is a little hard to understand.

The authors note that the presented experiments and dataset can be treated as a blueprint for further phenomena. While I see the transferability of the framework utilised, I wonder how diverse the use of a highly specialised corpus can be. This does not necessarily detract from the value of such corpora, but it would help if the authors elaborate on how future benchmarks should be built 'to detect additional layers of comprehension of natural languages'.

**Reproducibility:**

4: Could mostly reproduce the results, but there may be some variation because of sample variance or minor variations in their interpretation of the protocol or method.

**Reviewer Confidence:**

2: Willing to defend my evaluation, but it is fairly likely that I missed some details, didn't understand some central points, or can't be sure about the novelty of the work.

**Typos Grammar Style And Presentation Improvements:**

-Figure 5 is low resolution, resulting in a blurry image.
-line 389 Under ALT errors, for both clarity and consistency,  it would be great the authors give an example.

---

> ### Author Rebuttal · Authors · 2023-08-28
>
> The current dataset is one instance of a problem task that can be applied to any linguistic problem in any language.
> In this respect it is similar to the developments we have seen for example, in corpus annotation, for UD.
>
> There are many ways in which this approach can be extended. First, a specialised corpus on a given phenomenon in one language, might be explored for the very same phenomenon in another languages, especially for low-resource languages, given that the underlying linguistic phenomenon is the same.
>
> Secondly, templates for different problems can all be collected in a multi-problem dataset (we are currently exploring this testing method in another paper). Also, different templates for different problems can be merged in a single multi-problem template (we are also exploring this approach in a prompt-based setting).
>
> Finally, in comparison to other datasets, we think that the task itself is compelling and controllable, mostly thanks to the carefully crafted answer set which gives rise to controlled experiments.  The answer set supports error analyses whose variables are defined a piori, a proper experimental procedure, while other datasets instead  rely on error analyses that are done a posteriori. Our dataset leads to sound fine-grained investigations of large language models.

---

### Official Review · Reviewer_uWko · 2023-08-05

**Soundness:** 4

**Excitement:**

4: Strong: This paper deepens the understanding of some phenomenon or lowers the barriers to an existing research direction.

**Paper Topic And Main Contributions:**

This paper introduces a new benchmark dataset to help understand how sentence embeddings from pretrained language models capture complex properties of predicate-argument structures. The dataset follows the BLM (Blackbird’s Language Matrices) framework, and focuses on the spray/load verb alternations describing an event where an agent causes motions of a theme to a location, where both the theme and the location can function as the syntactic direct object of the verb. The task is to select one alternate from a set of answer options, given the other alternate and a sequence of context sentences indicating some shared properties of the argument structure.

The dataset is collected heuristically by first identifying similar verbs to spray and load from VerbNet, along with some example sentences extracted from the web, and then augmented with a masked language model combinatorially. Three types of datasets are constructed with increasing lexical difficulty, varying by whether the predicates and the arguments in the context and the answer set use the same lexical variant. The answer set is designed to exhibit different subpatterns of the argument structure that could be violated.

Experiments with several popular language models like RoBERTa and ELECTRA show that they all perform decently well on all three types of the dataset, achieving over 50% F1 even when trained on the easiest type and tested on the hardest. However, detailed error analysis shows that while the models manage to learn some aspects of the predicate-argument structures, there are other aspects that are still challenging, for example, understanding lexical semantic rules.

Main contributions:

a) Constructed a new benchmark dataset focusing on complex predicate-argument structures that can be used to study whether language models has a deep understanding of core syntactic and semantic properties of language.

b) Extensive experiments and detailed error analysis to show what type of knowledge the baseline models are capable of and what challenges still remain.


**Questions For The Authors:**

a) How do you construct the answer set? For some sub-patterns, it seems straightforward, but I’m not sure how to get NoEmb and LexPrep options from the heuristic process.

b) How do you select the 15k examples for Type 2 and 3? Is it by random sampling (as I understand the total possible combinations are much more than that)?

c) How do you get embeddings from RoBERTa or ELECTRA models? Is it from the first token, or mean pooling, or further trained with some recent sentence representation objectives (like SentenceBERT)? Also, do you keep the encoder weights frozen during training, or is it updated with the FFNN/CNN jointly?


**Reasons To Accept:**

a) The paper is well written and clearly explains the motivation and the complicated procedures to construct the dataset.

b) Comprehensive empirical results that might shed lights on future development of better models.


**Reasons To Reject:**

Some experimental details are not clearly stated or questionable, see my questions below.


**Reproducibility:**

3: Could reproduce the results with some difficulty. The settings of parameters are underspecified or subjectively determined; the training/evaluation data are not widely available.

**Reviewer Confidence:**

3: Pretty sure, but there's a chance I missed something. Although I have a good feel for this area in general, I did not carefully check the paper's details, e.g., the math, experimental design, or novelty.

**Typos Grammar Style And Presentation Improvements:**

a) Section 2.3, it might be helpful to include some examples for each subpattern that could be violated. Otherwise the readers will wait until Figure 7 to get a sense of what the incorrect answers look like, which also makes it hard to associate each answer example with their corresponding structures in Figure 3 and 4.

b) Figure 5 looks very blurry.

---

> ### Author Rebuttal · Authors · 2023-08-28
>
> We respectfully disagree that the data collection is performed "heuristically". We use naturally occurring examples as seeds for the automatic masked augmentation, which is then validated manually by expert linguists, a level of control that we believe is necessary.
> NoEmb and LexPrep represent errors with respect to the lexical preposition introducing the prepositional phrase (PP), but they aim to explore two different rules of the alternation. NoEmb aims to understand whether the PP is an argument and not a sub-phrase of the NP (e.g. the paint for the room). LexPrep aims to understand how much the lexicon, in particular the type of preposition, plays a role in the alternation. From the data creation perspective, LexPrep is generated by editing the PP with a preposition that does not appear in the original seed sentence (e.g. I spray the paint onto the wall; $=$ $\neg$ onto).  NoEmb follows the same rule but additionally involves a fill-mask task to retrieve lexical materials (e.g. 'the paint for the [MASK]'). All pertinent components are present in the lexical seed set, provided in the supplementary file.
>
> The type II and type III datasets were indeed randomly sampled.
>
> We use the embedding of the [CLS] (for BERT) and <s> (for RoBERTa/ Electra) token (which is token 0) from the output of the transformer. There is no additional processing on this token. The reason for using this is that we did not want to operate on something fine-tuned for a specific task, but rather we wanted to investigate the raw transformer output. There is no further tuning for the transformer. We use the pre-trained model to get the sentence embeddings. This is separate from what happens with the baselines (which are trained).
>
> If the paper is accepted, we will incorporate the reviewer's suggestions to improve the clarity of our presentation (e.g. an early presentation regarding the types of errors), taking advantage of the extra page.

---

### Official Review · Reviewer_fcXT · 2023-08-10

**Soundness:** 3

**Excitement:**

4: Strong: This paper deepens the understanding of some phenomenon or lowers the barriers to an existing research direction.

**Paper Topic And Main Contributions:**

This paper uses a recently introduced sentence pattern-matching task---Blackbird Language Matrices---to do detailed probing of language model understanding of a core syntax-semantics-interface phenomenon: the spray/load alternation.  The task consists of choosing a sentence which correctly continues a pattern (defined by different syntactic and semantic structures of arguments to a verb) from a set of alternatives which do not continue the pattern.

The paper introduces a rich dataset of instances of this task applied to the spray/load alternation (e.g. that "She sprayed the wall with the paint" and "She sprayed the paint on the wall" are both grammatical) and then probes RoBERTa and ELECTRA embeddings for their ability to complete the task.  The results suggest that the models are not bad at this task, but that they make a consistent type of error, usually relating more towards the semantics (e.g. not respecting animacy constraints) than to the syntax.

The data and results are both interesting and do seem to represent an advance both in what's known about these alternations in particular in language models and also in the methodology of linguistically-informed probing with the new task.  Because the task is so new, there are a few places where the paper could benefit from more detailed exposition and where more baseline comparisons would be helpful for knowing how to interpret the results we do see.  But I do suspect this paper to be of interest to many in the field of interpretability and analysis.

**Questions For The Authors:**

- Footnote 2 links to one sense of the verb _spray_ which is relevant.  Did you also use a sense of the verb _load_ and, if so, which?
- How exactly did you extract sentences in SPIKE (paragraph starting line 227)?  More details should be provided here or in an Appendix, in the interest of full reproducibility.
- Figure 7: "The contents were sticked in a box".  I think this should be "were stuck in a box", i.e. _stick_ has an irregular past participle. I doubt this makes a huge difference to anything, but am curious at which step the passivization occurs.
- Figures 9 and 10: can you clarify exactly what these plots are showing?  I _think_ it looks at all mistaken choices by the models, and shows which sentence was chosen instead of the correct one.  I think it would also be useful to have a real example mistake for each category; it was hard to determine what "ALT" meant at the bottom of the first column on page 6.  Relatedly, the discussion of ALT errors from lines 423 onward confused me a bit: are you saying that the model tends to select sentences of these two types?  The wording almost makes it sound like it's generating those, but I don't think that's the task.

**Reasons To Accept:**

- Application of a new diagnostic task method (Blackbird's language matrices) to a subtle linguistic phenomenon (spray/load alternation) at the syntax-semantics interface.  This seems like a very valuable contribution, showing that we can do linguistically motivated analysis work in a richer setting than normal classification-type tasks.
- Detailed error analysis of models, showing that e.g. they suffer with selectional restrictions on arguments (animacy).  The main results and this analysis refine earlier probing work on verb alternations.
- Data will be made publicly available when non-anonymized; I can imagine this being used by other researchers.

**Reasons To Reject:**

- Some details of dataset construction were not described in sufficient detail.
- Lack of baseline comparisons for the BLM task / dataset.  There are at least three things that would be welcome here: What's random chance performance on the task?  How well does a simple baseline (e.g. bag of word2vec vectors) do on the task?  Most crucially, but most difficult: how well do _people_ do on this task?  Just working through the examples, I personally find the task fairly difficult.  It's not obvious to me that we should expect a normal native speaker of English to get 100% on this task; it would be really useful to know how well people do, as well as what kinds of errors they make.  (The authors acknowledge the lack of human comparison in the limitations section.)
- It would be nice to evaluate some _large_ language models in a prompting-based setting both as a comparison and to show that the BLM method can be used with models of that type.

**Reproducibility:**

3: Could reproduce the results with some difficulty. The settings of parameters are underspecified or subjectively determined; the training/evaluation data are not widely available.

**Reviewer Confidence:**

3: Pretty sure, but there's a chance I missed something. Although I have a good feel for this area in general, I did not carefully check the paper's details, e.g., the math, experimental design, or novelty.

**Typos Grammar Style And Presentation Improvements:**

- In the paragraph first illustrating the spray/load alternation (line 86 and following, page 2), colors are used for the different semantic roles in the two sentences.  I think I would also similarly color-code the names "Agent", "Theme", and "Location", just to make the parallel especially clear.
- An example of a BLM from earlier work in Section 2 (i.e. before 2.1) would be welcome, since this is a relatively new task that will be unfamiliar to many readers.
- Similarly, I think Figures 2 and 3 would benefit a lot from additionally being instantiated with examples of the template, e.g. using the example alternation from the Introduction.  (These examples come up as separate figures 6 and 7, but I wanted to see those earlier.)
- The open parenthesis on line 154 has no closing parenthesis.
- Figure 5 has quite low resolution.
- Line 346: I'm not familiar with the $[ \cdot ]^+$ notation, and would personally prefer $\max (0, \dots)$.
- Section 4.2: "Figure 13" I think should be "Figure 8".  Also, this paragraph says "left handside" and "right handside" for what I think should be "top row" and "bottom row".

---

> ### Author Rebuttal · Authors · 2023-08-28
>
> The reviewer asks about data extraction.  We clarify here the extraction of the seed examples from SPIKE. We performed the search of different inflected forms of the verbs and analysed manually the naturally occurring examples in the output. We removed non-target examples (e.g. the PP does not represent an argument of the verb, but it is  embedded in the nominal, as in 'This bulb scatters dim light with a green hue around' (ID: AMG Enthusiast; AI9JTX5ZX0OJG)). Care was also taken to vary the types of sentences as much as possible (e.g. imperatives, relatives) and pronominal entities (e.g., to counter the fact that a good portion of Amazon reviews are in the 1st person singular). In the final version, we will take advantage of the extra page to explain in detail and integrate additional information, such as the enrichment of the discussion on the link of footnote 2.
>
> We respectfully disagree that we do not have baseline comparisons. This paper concentrates on the description of the data and its benchmarking with basic architectures. We think this work represents a baseline in and of itself, the architectures we propose are standard, simple and do not integrate any novel insights or complex computational mechanisms. We also do not claim any preference for one architecture over another.
>
> In work described in a different paper under submission, we compare these baseline generic architectures to more tailored and more articulate VAEs and do indeed show an improvement on several BLM problems.
>
> As for a random baseline on this dataset, the task is multiple-choice, where, by the construction clearly described in the paper, 9 equally frequent options are available in the answer set, then the random baseline is 1 over 9.
>
> As for comparison to humans, this is indeed planned work but it requires a whole dedicated paper due to the nature of the task. This task is not a natural task that all speakers will perform equally well. This is analogous to an IQ task, Raven's progressive matrices. The output from the psychological literature on RPMs shows that problem-solvers fall into distinct groups, with some achieving high scores and others achieving lower. This means that the establishing human performance needs a correlation with some other measure of abilities and clear control of the constraints in the experimental procedure (size of the dataset, carry-over effects, priming effects from easier tasks).
>
> We have performed informal tests on BLMs. They show that speakers easily grasp type I, can still see the patterns in type II data, but find type III hard.
>
> As for prompt-based settings, we are currently exploring prompt-based settings in a different paper. Again, given the nature of the task, a detailed qualitative evaluation is needed to establish if, but mostly how large language models solve the BLM task. Initial results show that, decomposing the solution of this task into the subskills that humans have been argued to deploy, large language models do well in the individuating subtasks (identifying the objects to manipulate and their attributes), but do not do so well in the integrative substasks that allow to eventually solve the task.
>
> All lexical materials, among which irregular past tense, have been manually reviewed and corrected when needed. We will verify again the validation. Passivization is automatically created during the template constructions, since dedicated lexical entries of the lexical seed set involve the NPs associated with the past tense of the verb 'to be' and the past tense of the verb under investigation.
>
> The plots in Figure 9 and 10 show the error analysis of our models, in terms of distributions. We added examples in the running text (we acknowledge that the examples for ALT are located relatively late in the previous manuscript, lines 426-431), but we can add them in the caption if needed. In lines 423-onwards, we comment on the fact that ALT represents an error that occurs more in type III than in the other two types, across templates, but it is always based on a multiple-choice task and not based on a generation task.
>
> Finally, the presentation improvements proposed by the reviewer and correction of the typos will be taken into account in the new version of the manuscript.

---

### Meta-Review · Area_Chair_G4JP · 2023-09-14

**Recommendation:** 4

**Metareview:**

This paper probes sentence embeddings from two pretrained language models, RoBERTa and ELECTRA, to better understand how well these models capture and generalize properties of spray/load verb alternations in English. It uses a sentence pattern-matching task (Blackbird Language Matrices) to do so, and shares the dataset and findings from resulting experiments, which suggest the models are adequate at the task but commit a consistent type of error. All in all, reviewers praised the novelty of the task and the leveraging of linguistic insights to investigate model “understanding”, as well as the detailed qualitative error analysis. There was some concern, however, that experimental and dataset construction details were missing and unclear (e.g. how were the embeddings acquired from RoBERTa and ELECTRA, how were sentences extracted in SPIKE, Figures 2, 3, 9, 10 could use clarifying details, etc.). One reviewer also requested a more detailed exposition of the maintenance and extensibility of this dataset to other phenomena. Lastly, there was some disagreement over the inclusion of baseline comparisons for this task, e.g. random chance, bag of word2vec vectors, human performance. The authors defended their choice to exclude these comparisons, and while I find their rebuttal satisfactory, it would perhaps be beneficial to include the points raised in the rebuttal in the revised paper.

In summary, quite a few minor revisions are needed to address all reviewers’ comments and concerns, which would benefit the Soundness (two 3s, one 4) and clarity of the paper. Barring one reviewer, the Excitement scores were quite high (two 4s, one 2); I would advise the authors to consider the concerns of that reviewer though, i.e. the maintenance and extensibility of the dataset. Given how novel and specific the task is, understanding how it contributes to the wider field of language model analysis and interpretability would raise the impact and interest of this paper.

---

### Decision · Program_Chairs · 2023-10-07

**Decision:**

Accept-Findings

**Comment:**

This paper probes sentence embeddings from two pretrained language models, RoBERTa and ELECTRA, to better understand how well these models capture and generalize properties of spray/load verb alternations in English. It uses a sentence pattern-matching task (Blackbird Language Matrices) to do so, and shares the dataset and findings from resulting experiments, which suggest the models are adequate at the task but commit a consistent type of error. All in all, reviewers praised the novelty of the task and the leveraging of linguistic insights to investigate model “understanding”, as well as the detailed qualitative error analysis. There was some concern, however, that experimental and dataset construction details were missing and unclear (e.g. how were the embeddings acquired from RoBERTa and ELECTRA, how were sentences extracted in SPIKE, Figures 2, 3, 9, 10 could use clarifying details, etc.). One reviewer also requested a more detailed exposition of the maintenance and extensibility of this dataset to other phenomena. Lastly, there was some disagreement over the inclusion of baseline comparisons for this task, e.g. random chance, bag of word2vec vectors, human performance. The authors defended their choice to exclude these comparisons, and while I find their rebuttal satisfactory, it would perhaps be beneficial to include the points raised in the rebuttal in the revised paper.

In summary, quite a few minor revisions are needed to address all reviewers’ comments and concerns, which would benefit the Soundness (two 3s, one 4) and clarity of the paper. Barring one reviewer, the Excitement scores were quite high (two 4s, one 2); I would advise the authors to consider the concerns of that reviewer though, i.e. the maintenance and extensibility of the dataset. Given how novel and specific the task is, understanding how it contributes to the wider field of language model analysis and interpretability would raise the impact and interest of this paper.